# VIDEO PANELS FOR LONG VIDEO UNDERSTANDING

## ABSTRACT

Recent Video-Language Models (VLMs) achieve promising results on long-video understanding, but their performance still lags behind that achieved on tasks involving images or short videos. This has led to great interest in improving the long context modeling of VLMs by introducing novel modules and additional complexity. In this paper, we take a different approach: rather than fine-tuning VLMs with the limited data available, we attempt to maximize the performance of existing models. To this end, we propose a novel visual prompting strategy specifically designed for long-video understanding. By combining multiple frames as panels into one image, we effectively trade off spatial details for temporal resolution. Our approach is training-free, parameter-free, and model-agnostic, and can be seamlessly integrated into existing VLMs. Extensive experiments on five established benchmarks across a wide range of model architectures, sizes, and context windows confirm the consistency of our approach. For the TimeScope (Long) dataset, which has the longest videos, the accuracy for video question answering is improved by up to 19.4%. Overall, our method raises the bar for long video understanding models. We will make our code available upon acceptance.

## 1 INTRODUCTION

Connecting the reasoning capabilities of Large Language Models (LLMs) with other modalities has opened up the possibility to reason about and interact with multi-modal data (Yin et al., 2024; Yi et al., 2025). Vision–Language Models in particular, which are capable of reasoning over images or videos based on a textual query, now have impressive perceptive and cognitive capabilities. This has led to their application across a variety of imaging domains like medical imaging (Hartsock & Rasool, 2024) and industrial defect detection (Mokhtar et al., 2025), and progress is further driven by the publication of strong open-source models such as LLaVA-Video (Zhang et al., 2025b), LLaVA-OneVision (Li et al., 2024b), and Qwen2.5-VL (Bai et al., 2025). However, with image datasets and benchmarks far outnumbering those for videos, video understanding by Video–Language Models (VLMs) remains a challenge.

While VLMs can still achieve high performance on tasks involving short video clips, their performance drastically decreases as video length increases. For example, Qwen-VL2.5 exhibits a notable drop in accuracy when processing videos longer than three minutes (Zohar et al., 2025). This decline is largely caused by their limited temporal resolution, stemming from relatively small context windows and memory constraints, which restrict the effective visual context that the model can process. Nonetheless, the importance of reasoning about long videos is crucial for many tasks, setting long-video understanding as one of the crucial challenges in VLM research.

Recent works try to improve the long context modeling of VLMs in a variety of ways, typically by heavily compressing the tokens used to represent input frames before modeling the temporal dynamics (Bai et al., 2025; Li et al., 2024d; Zhang et al., 2025a). Although recent approaches add new modules, additional training procedures, and additional training data to a base model, which makes the training more complex, they struggle to consistently outperform the base models with limited context sizes (Shu et al., 2025; Wang et al., 2025a; Zhang et al., 2024b). We therefore address the question: *Is there a simple yet efficient way to consistently improve long video understanding of existing VLMs?*

In this paper, we thus take a different approach. Rather than fine-tuning VLMs for long-video understanding with the limited instruction data available, we focus on making the most out of the capabilities of existing models. To this end, we introduce the first visual prompting approach designed

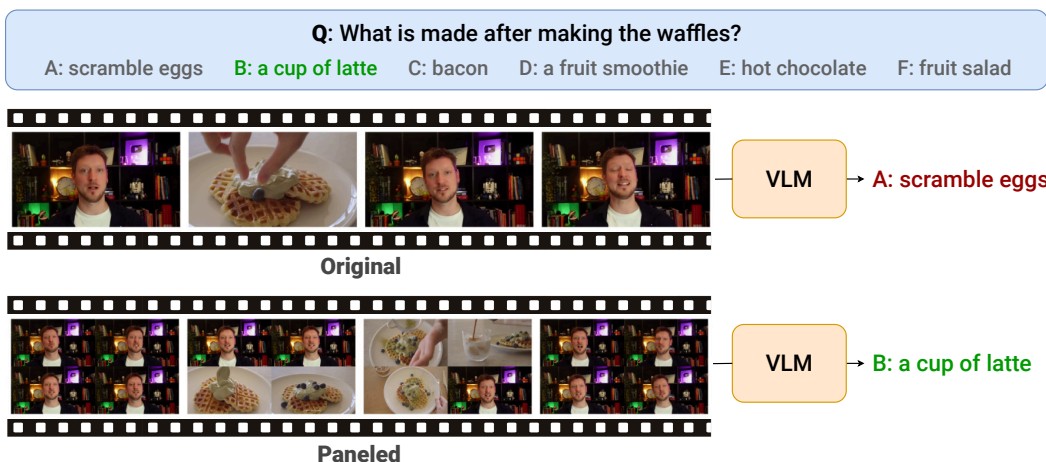

Figure 1: **Illustration of our approach.** We show the output of LLaVA-OneVision 7B on a sample from Video-MME. Without our proposed visual prompting, the model is unable to answer the question correctly. By paneling multiple frames into one, we increase the capacity for long video understanding and output the correct answer, without introducing any additional complexity.

specifically for long video understanding. Specifically, we combine multiple frames of a video into one image, as panels in a comic as shown in Fig. 1. Our method leverages these multi-panel inputs to improve temporal reasoning: a sequence of such panels enables denser temporal coverage within the model's constrained input window. This effectively increases temporal resolution at the cost of some spatial detail, leading to a better balance between them for long videos. Since our approach does not modify the underlying VLM architecture and is training-free and model-agnostic, it is a versatile approach that can be applied to any existing VLM. Additionally, we show that fine-tuning these models with their original, short-video training data represented in panels further amplifies performance gains on long videos.

We evaluate our approach on 5 video question answering benchmarks, namely VideoMME (Fu et al., 2025), TimeScope (Zohar et al., 2025), MLVU (Zhou et al., 2025), MF2 (Zaranis et al., 2025), and VNBench (Zhao et al., 2024), and apply our approach to 7 VLMs, namely Video-LLaVA (Lin et al., 2023), VideoChat2-HD (Li et al., 2024c), LLaVA-OV (Li et al., 2024b), Qwen-2VL (Wang et al., 2024), Qwen-2.5VL (Bai et al., 2025), LLaVA-Video (Zhang et al., 2025b), VideoLLaMA 3 (Zhang et al., 2024b), which use between 8 and 180 context frames. Our approach improves the base VLMs in most settings and on average over all datasets for all VLMs. For the TimeScope (Long) dataset, which has the longest videos, the accuracy of VideoLLaMA 3, which is the strongest base model on this dataset, is improved by a large margin. The accuracy is increased by $+7.6$, which is an improvement by $19.4\%$. This shows that paneled images are a simple yet efficient approach to improve the long video understanding performance of VLMs.

Overall, the main contributions of this paper are as follows:

- We present the first visual prompt engineering method for long-video understanding that requires no training or additional parameters, and can be seamlessly integrated into existing VLMs.
- We show through extensive experiments that our framework improves results across a wide variety of benchmarks, model types, and context sizes.
- We show that fine-tuning can increase the performance even further, without introducing new training data.

## 2 RELATED WORK

**Long Video Understanding.** Scaling vision–language models (VLMs) to long videos is challenging due to the large number of visual tokens and the limited context windows of language backbones.

A common strategy is to reduce the input size, either by modifying visual re-samplers to extract fewer tokens (Li et al., 2023; 2024e; Cheng et al., 2024) or by pruning and merging features through heuristic techniques (Chen et al., 2024; Shang et al., 2024; Jin et al., 2024; Zhou et al., 2024). Other works extend the sequence capacity of the LLM, as in LongVA (Zhang et al., 2024b), who transfer the long-context pretraining of their language backbones to multi-modal settings. Another line of work aims to preserve temporal coherence through memory mechanisms or summarization tokens. For instance, VideoLLaMB (Wang et al., 2025a) introduces recurrent memory bridges with scene segmentation, while Video-XL (Shu et al., 2025) condenses video intervals into summarization tokens within the Transformer. These divergent strategies highlight the difficulty in finding a good balance between efficiency and information preservation, and motivate the exploration of lightweight alternatives such as prompt-based strategies.

**Visual Prompting in VLMs.** Visual prompting, or visual prompt engineering, is a strategy that modifies the visual input (images or videos) of VLMs to steer their attention, rather than altering the model itself. Prior studies show that even simple geometric cues, such as e.g., colored regions (Yao et al., 2024) or red circles (Shtedritski et al., 2023) for CLIP (Radford et al., 2021), or bounding boxes (Dang et al., 2023; Duan et al., 2024; Ma et al., 2024), can effectively guide focus in image understanding tasks. In multi-modal settings, visual prompting has been successfully applied to improve alignment and reasoning in MLLMs (Cai et al., 2024; Wu et al., 2024). Extending this idea to videos, Wu et al. (2025) introduce numerical tags as visual prompts within video frames, Du et al. (2025) improve fine-grained motion recognition with motion blur and spotlighting, and Wang et al. (2025b) improve emotion recognition with an ensemble of visual prompting techniques. Our work follows a related intuition: we project temporal information into the spatial domain by concatenating subsequent frames into a single visual input. This design leverages visual prompting as a lightweight yet effective way to enhance long-video understanding without introducing additional complexity.

**Composite Images.** Combining multiple images into a single representation has been explored in several domains, such as multi-sensor fusion for autonomous driving (Li et al., 2024b), action recognition (Fan et al., 2022), efficient retrieval (Nishimura et al., 2024), and comic book understanding (Iyyer et al., 2017). These works demonstrate the effectiveness of spatially arranging visual inputs to capture relationships across multiple sources or instances. However, we are the first to apply this idea to long video understanding.

## 3 METHOD

We focus on improving video understanding of VLMs as measured by their question answering accuracy. For this scenario, we make use of datasets in the form of $\{\mathbf{x}_i, q_i, y_i\}_{i=1}^N$ with videos $\mathbf{x} \in \mathbb{R}^{D \times 3 \times H \times W}$ of duration D, multiple-choice questions $q \in \Sigma^*$ from an alphabet $\Sigma$, and correct answers $y$. The VLM is typically soft-constrained by a prompt to only output the letter of the correct answer, allowing for an easy evaluation and comparison between methods by measuring their accuracy (Bai et al., 2025; Fu et al., 2025).

### 3.1 MOTIVATION

Video understanding aims to determine the extent to which VLMs can reason about the videos they receive as input. While for short videos, this often comes in the form of action recognition, longer videos allow for higher-level questions. This makes them especially interesting, as reasoning aspects such as event ordering and long-range dependencies can now be considered. However, this increase in the sophistication of the questions comes with a notable increase in difficulty.

The main challenge is the limited temporal resolution of the VLMs, which is defined by their *context window* $C$ and determines how many frames (or tokens) the model can process. In the case of long videos, i.e., the case where $D \gg C$, this means that the model can not densely parse the entire input video. Instead, the temporal resolution has to be reduced in order to fit within this constraint. In practice, this is done by a sampling function $\phi : \mathbb{R}^{D \times 3 \times H \times W} \to \mathbb{R}^{T \times 3 \times H \times W}$ that samples $T$ frames from the input video.

The application of $\phi$ to long videos creates an important imbalance. VLMs are largely trained on images and short videos, where the spatial and temporal resolution are both still high after sampling.

For long videos, however, the temporal resolution decreases drastically, while the spatial resolution remains the same. This mismatch leads to a disproportionately large part of the computational power of the model being dedicated to spatial, rather than temporal, relations.

## 3.2 Visual Prompting for Long Video Understanding

To re-balance the allocation of computational resources between the spatial and temporal components, we introduce the first visual prompting technique for long video understanding by combining multiple frames into a single image. By creating a sequence of these multi-panel frames, we enable denser temporal coverage within the given input budget, thereby increasing temporal resolution at the expense of some spatial detail. Beyond simply seeing more frames, this allows the model to use its strong pre-trained visual encoder to infer temporal relations as well. In this way, our approach effectively extends the video context of existing VLMs. Our method operates in two steps:

**Dynamic frame sampling.** The requirements for understanding short and long videos differ (Li et al., 2024d), and our prompting strategy is designed explicitly for long videos. Therefore, we choose the number of sampled frames $T$ dynamically. We set $T$ depending on the ratio between the context window size $C$ and the duration $D$,

$$T = \begin{cases} C, & \text{if } \gamma C \geq D, \\ \alpha\beta C, & \text{otherwise.} \end{cases} \tag{1}$$

As a result, we only use paneling in the case where the sampled frames are spaced at least $\gamma$ frames apart. The hyperparameters $\alpha$ and $\beta$ define the number of frames that are combined horizontally and vertically, respectively, into one image. We find that our method reaches the best results when $\alpha=\beta$, and, therefore, these can be considered as a single hyperparameter. Fig. 1 shows our default setting with $\alpha=\beta=2$. We explore other combinations of the hyperparameters in the ablation study.

**Panel construction.** In the case where $\gamma C \geq D$, the resulting sampled video $\mathbf{x} \in \mathbb{R}^{\alpha\beta C \times 3 \times H \times W}$ is too large for the context window of the VLM. Therefore, we downsample $\mathbf{x}$ to $\mathbf{x}' \in \mathbb{R}^{\alpha\beta C \times 3 \times H/\alpha \times W/\beta}$. Then, every $\alpha\beta$ frames are stacked into one image in left-to-right, top-to-bottom order into a *panel image*. This way, the final input to the VLM, $\mathbf{x}'' \in \mathbb{R}^{C \times 3 \times H \times W}$, is again within the constraints expected by the vision encoder. For example, in the case of $\alpha=\beta=2$, we have:

$$\mathbf{x}''_i = \begin{pmatrix} \mathbf{x}'_{4i} & \mathbf{x}'_{4i+1} \\ \mathbf{x}'_{4i+2} & \mathbf{x}'_{4i+3} \end{pmatrix}.$$

This procedure preserves the standard input shape while extending temporal coverage by a factor of $\alpha\beta$.

We found that for specific models, describing the panels in the prompt improved results. However, we did not find a single textual prompt that increases performance across models uniformly. As a result, we do not provide any extra information in the prompt. Nonetheless, if the model architecture is fixed, an appropriate description can improve results further. We explore this further in the Appendix.

## 3.3 Fine-tuning

Applying our approach to existing VLMs improves zero-shot performance on long-video benchmarks, even though these models are originally trained on standard videos without panels. When additional resources are available, VLMs can be fine-tuned to better adapt to the proposed input format and further enhance performance. Specifically, we fine-tune the models on the video-question pairs from their original training data by maximizing the likelihood of the correct multiple-choice answer:

$$\ell_{FT}(\mathbf{x}, q, y) = -\log p_\theta(y \mid \mathbf{x}, q). \tag{2}$$

We discuss the impact of fine-tuning in Section 4.3.

## 4 Experiments

We perform experiments across five well-established datasets and models to evaluate our visual prompting strategy and show that it is model-agnostic and generally applicable.

### 4.1 EXPERIMENTAL SET-UP

**Models.** Since our approach works on the input level and does not need any architecture changes, we can apply it directly to a large variety of existing VLMs. In particular, we focus on three main groups of VLMs:

- **Small-context.** VLMs taking up to 16 frames as input. We evaluate Video-LLaVA (Lin et al., 2023) and VideoChat2-HD (Li et al., 2024c).
- **Medium-context.** 16-128 frames as input. We evaluate three variants of LLaVA-OV (Li et al., 2024b) (0.5, 7, and 72 billion parameter), Qwen-2.5VL (Bai et al., 2025) with 32 input frames, and LLaVA-Video (Zhang et al., 2025b) in two variants (7 and 72 billion).
- **Long-context.** VLMs taking more than 128 frames as input. We evaluate Qwen-2VL (Wang et al., 2024), Qwen-2.5VL (Bai et al., 2025), and VideoLLaMA 3 (Zhang et al., 2024b), all with 180 frames input.

**Datasets.** We use six established and public benchmarks for our evaluation:

- **VideoMME** (Fu et al., 2025) (VMME). 2,7000 videos between 11 seconds and 1 hour long, categorized as *short* videos (80 seconds on average), *medium* (500 seconds) and *long* (2500 seconds).
- **TimeScope** (Zohar et al., 2025). Videos are divided into 13 different lengths, ranging from 60 to 36,000 seconds. We divide the evaluation into `short` (up to 3 hours, average 2590 seconds, 1500 videos) and `long` (videos lasting 5, 8, or 10 hours, average 27,600, 450 videos).
- **MLVU** (Zhou et al., 2025) covers video lengths from 3 minutes to 2 hours, with an average length of 15 minutes. We evaluate on the *dev* set, which consists of 2593 QA pairs over 1730 videos.
- **MF2** (Zaranis et al., 2025) uses full-length movies with an average duration of 88.3 minutes. The task is to discriminate between true and false claims across 850 claim-pairs.
- **VNBench** (Zhao et al., 2024) (VNB) consists of 5400 questions on videos ranging from 10 to 180 seconds

Extended details on the datasets are provided in Sec. B. We report the accuracy over all the questions for each benchmark.

**Implementation Details.** We evaluate all models with `lmms-eval` (Li et al., 2024a; Zhang et al., 2024a). The VLMs are given the question and the multiple-choice answers in the format expected by the model, after which all VLMs get the standard instruction to "Answer with the option's letter from the given choices directly.\n". We set $\alpha = \beta = 2$, $\gamma$ to frames per second of the input video, and use uniform sampling for $\phi$.

For fine-tuning, we optimize LLaVA-OneVision 7B on the video subset of LLaVA-Video-178K (Zhang et al., 2025b) for one epoch, using a batch size of 2 and a gradient accumulation step of 4.

### 4.2 MAIN RESULTS

We show in Table 1 that our paneling approach consistently improves upon the baseline across a wide range of model architectures, parameter counts, context windows, and datasets, and using panels is, on average, the best approach for all models tested. The benefits are consistent across question types, but are especially pronounced in datasets where the task is to reason about "needles", i.e., specific moments inserted in a video, such as TimeScope or VNBench. For example, paneling raises the VNBench score of LLaVA-OneVision 72B by 3.2 percentage points, and the VMME and long TimeScope scores are raised by 2.0 and 1.3 points, respectively, for Qwen-2.5 VL with 32 context frames. For VideoLLaMA 3 7B, our approach even raise the accuracy from 39.1 to 46.7, which is an increase by 7.6 points or 19.4%. There are some cases where we outperform previously reported numbers. For instance, our long-context settings for Qwen2.5-VL are slightly higher than those reported on, e.g., VideoMME (Bai et al., 2025) (65.1 versus 66.0). Even there, our panels

| | #frames | VMME | | | TIMESCOPE | | MLVU | MF2 | VNBench | Avg. |
|---|---|---|---|---|---|---|---|---|---|---|
| | | medium | long | overall | Short | Long | | | | |
| Avg. Duration (s) | | 516 | 2467 | 1018 | 2586 | 27600 | 651 | 5300 | 57 | |
| *Small-context VLMs* | | | | | | | | | | |
| Video-LLaVA 7B | 8 | 36.6 | 32.6 | 37.1 | 24.4 | **17.6** | **45.7** | **50.4** | 27.8 | 33.8 |
| + ours | 8 | **37.9** | **34.2** | **38.7** | **25.6** | 17.1 | **45.7** | 50.2 | **32.0** | **34.8** (+1.0) |
| VideoChat2-HD | 16 | 26.4 | 24.8 | 25.2 | 21.2 | **19.8** | 49.2 | **50.0** | 27.9 | 32.2 |
| + ours | 16 | **26.7** | **25.1** | **25.4** | **21.3** | **19.8** | **49.8** | **50.0** | **28.5** | **32.5** (+0.3) |
| *Medium-context VLMs* | | | | | | | | | | |
| LLaVA-OV 0.5B | 32 | 40.9 | **37.0** | 43.8 | 49.4 | 25.6 | **44.0** | 50.1 | 39.8 | 42.1 |
| + ours | 32 | **42.6** | 36.6 | **44.3** | **56.9** | **30.0** | 43.1 | **50.2** | **41.0** | **44.3** (+1.2) |
| LLaVA-OV 7B | 32 | **56.7** | 48.8 | 58.5 | 58.7 | 30.2 | 62.9 | 51.5 | 54.8 | 52.8 |
| + ours | 32 | 56.2 | **50.2** | **58.9** | **69.5** | **33.8** | **65.3** | **52.1** | **57.7** | **56.2** (+3.4) |
| LLaVA-OV 72B | 32 | 62.9 | 57.6 | 66.0 | 59.1 | **33.8** | 21.6 | 56.6 | 59.4 | 49.4 |
| + ours | 32 | **66.4** | **59.3** | **67.7** | **70.0** | 32.4 | **23.8** | **58.5** | **62.6** | **52.5** (+3.1) |
| Qwen-2.5VL 7B | 32 | 61.6 | 51.2 | 61.9 | 52.8 | 28.7 | 60.1 | 52.6 | 55.6 | 51.9 |
| + ours | 32 | **64.0** | **54.7** | **63.9** | **60.8** | **30.0** | **64.9** | **53.8** | **58.5** | **55.3** (+3.4) |
| LLaVA-Video 7B | 64 | **62.3** | 53.6 | 64.3 | 64.8 | 34.7 | **66.2** | 52.8 | - | 56.6 |
| + ours | 64 | 62.2 | **54.0** | **64.4** | **79.2** | **39.3** | 66.0 | **54.4** | - | **60.7** (+4.1) |
| LLaVA-Video 72B | 64 | 67.8 | 61.2 | 69.8 | 65.4 | **30.9** | 52.9 | 58.2 | - | 55.4 |
| + ours | 64 | **68.6** | **61.4** | **70.1** | **75.7** | **30.9** | **54.4** | **59.9** | - | **58.2** (+2.8) |
| *Long-context VLMs* | | | | | | | | | | |
| Qwen-2VL 7B | 180 | 62.7 | 51.0 | 62.4 | 66.1 | 23.8 | 65.7 | 54.3 | - | 54.5 |
| + ours | 180 | **62.9** | **52.7** | **63.0** | **71.2** | **26.7** | **65.8** | **56.7** | - | **56.7** (+2.2) |
| Qwen-2.5VL 7B | 180 | **67.6** | 54.8 | 66.0 | 73.9 | **37.6** | 66.7 | 54.2 | - | 59.7 |
| + ours | 180 | 66.9 | **56.1** | **66.3** | **79.1** | 35.6 | **66.8** | **54.8** | - | **60.5** (+0.8) |
| VideoLLaMA 3 7B | 180 | **64.6** | 54.1 | 65.3 | 80.2 | 39.1 | **47.3** | **58.9** | - | 58.2 |
| + ours | 180 | 63.7 | **55.1** | 65.3 | **87.2** | **46.7** | 47.1 | 58.3 | - | **60.9** (+2.7) |

Table 1: **Benefits of using our approach across datasets and models.** We report the video question answering accuracy. The overall set of VMME, report Since the videos of VNBench are very short, we evaluate only VLMs with up to 32 frames context on it. Our method improves the baseline results in nearly all cases and in average for all VLMs.

improve results further to 66.3, showing that even with close to optimal settings, we can increase performance.

Nonetheless, there are still some cases where paneling slightly hurts the results, such as the performance of VideoLLaMA 3 7B on MF2 dropping by 0.6 points, and some other isolated drops in performance can be found. However, these cases do not occur often and are to be expected with such a wide evaluation.

If we compare the averages over the datasets for each VLM, we observe a consistent improvement. For the VLMs with small context, i.e., 8-16 frames, the average improvement is between 0.3 and 1.0 points. For the medium-context VLM, the average improvement is even higher and between 1.2 and 4.1 points. For the VLMs with 180 frames as context, the gain is between 0.8 and 2.7 points. It is interesting to see that our approach with LLaVA-Video 7B, which uses 64 context frames, outperforms the long-context base VLMs. If we add our approach to the long-context VLMs, they remain the best performing approaches but the gap is much smaller (60.9 for VideoLLaMA 3 7B vs. 60.7 LLaVA-Video 7B). This shows on the one hand that our approach is versatile and improves various VLMs for video question answering without any additional training or increase of parameters. On the other hand, it questions whether existing VLMs efficiently utilize the larger context.

## 4.3 FINE-TUNING

We show the results for fine-tuning with and without the proposed paneling in Table 2 to disentangle the effect of paneling from the fine-tuning itself. We find that fine-tuning further boosts performance by 0.4 and 0.6 on VMME and long TimeScope, respectively, over its zero-shot application when fine-tuning the projection and LLM modules. In the case without paneling, the average performance on VideoMME remains unchanged, while for Timescope, a slight improvement for long videos, but an equivalent drop for short videos, yields no net benefit. Overall, these results show that paneling images is an effective representation for long videos in general, both in the zero-shot setting and with fine-tuning.

|  | VMME | | | | TIMESCOPE | |
|---|---|---|---|---|---|---|
|  | short | medium | long | overall | Short | Long |
| *Without Panels* | | | | | | |
| No Fine-Tuning | 70.1 | 56.7 | 48.8 | 58.5 | 58.7 | 30.2 |
| Proj + LLM | 71.1 | 56.2 | 48.2 | 58.5 | 58.0 | 30.9 |
| *With Panels* | | | | | | |
| No Fine-Tuning | 70.1 | 56.2 | 50.2 | 58.9 | 69.5 | 33.8 |
| Proj + LLM | 70.4 | 58.1 | 49.4 | 59.3 | 69.5 | 34.4 |

Table 2: **Effect of fine-tuning.** We report results on VMME using LLaVA-OV 7B. *Proj* is the projector between the vision encoder and the Large-Language Model (LLM). Fine-tuning with panels reaches superior performance over fine-tuning with standard inputs.

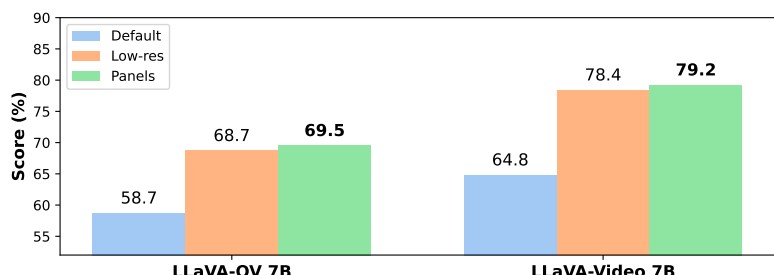

Figure 2: **Comparison to token reduction baseline.** We show results for LLaVA-OneVision 7B and LLaVA-Video 7B on TimeScope with three different reduction strategies: default (no reduction), *low-res*, and panels. Panels consistently improve the results.

### 4.4 WHY PANELS?

Our approach provides a natural way to spend more tokens on temporal rather than spatial resolution. While our approach is model-agnostic, as the visual prompting is done at the input level, there are other ways to do so. To further justify the choice for visual prompting with panels, we compare our method against another way of increasing the ratio of tokens used for temporal understanding.

We perform the analysis with LLaVA-OneVision and LLaVA-Video. We use average pooling on the visual tokens right before projection into the LLM. As there are $27 \times 27$ tokens per image at this point, we zero-pad to $28 \times 28$ before pooling. This way, this baseline uses slightly more tokens than our proposed panels (25088 vs. 23328). We refer to this baseline in both cases as *low-res*. For a fair comparison, we only apply *low-res* when the videos are at least $\gamma C$ frames long (Eq. 1) and use normal sampling otherwise, similar to our own method.

We show the results of our comparison in Fig. 2. Our strategy outperforms both the default strategy and *low-res* for all the models, except for LLaVA-Video 7B on TimeScope, where *low-res* reaches equal performance. Overall, these results show that our approach is the optimal way of increasing temporal resolution and effectively allows VLMs to use context sizes far bigger than what they were trained with.

Our findings are in line with previous works on high-resolution image understanding with VLMs, which found that scaling resolution is more effective than scaling the number of tokens (Li et al., 2024b; Liu et al., 2024). Our approach can be seen as the inverse of this strategy; rather than splitting one high-resolution image into multiple chunks, we combine many high-resolution frames into one image, outperforming approaches based on token pooling.

### 4.5 ABLATIONS

**Performance for different video lengths.** TimeScope (Zohar et al., 2025) provides videos with corresponding question–answer pairs across 13 different video lengths, allowing us to analyze how

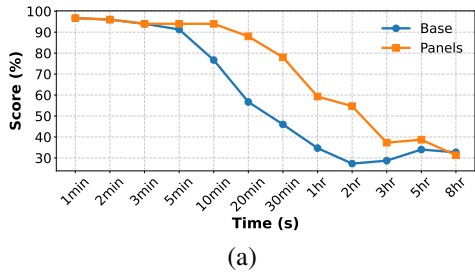 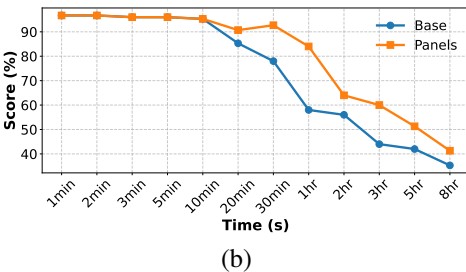

(a)                  (b)

Figure 3: **Performance as a function of video duration.** We show results for (a) LLaVA-Video 7B and (b) Video-LLaMA 7B on short and long TimeScope. Panels consistently improve the results for longer videos.

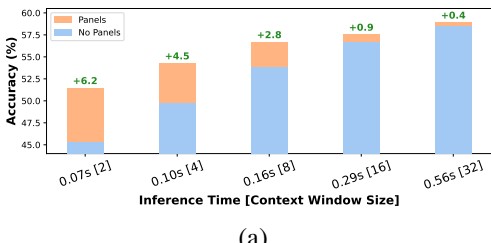 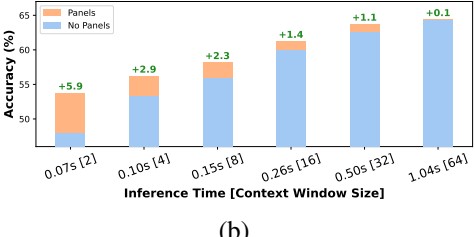

(a)                  (b)

Figure 4: **Performance as a function of context window size.** We show results for (a) LLaVA-OneVision 7B and (b) LLaVA-Video 7B. Panels consistently improve the results, with more pronounced effects for smaller window sizes.

paneling affects performance as duration increases. The results for LLaVA-Video 7B and Video-LLaMA 7B in Fig. 3 show that our visual prompting strategy consistently improves accuracy on longer videos, confirming the efficacy of paneling inputs.

**Trade-off between accuracy and inference time.** The same VLM can process different numbers of frames as input. Using fewer frames reduces the number of visual tokens to process, which in turn lowers inference time and computational cost. In Fig. 4, we show how paneling impacts performance across a range of different context window sizes, alongside the relative inference time. Our prompting strategy is effective for all window sizes. Furthermore, the gain in performance allows for fewer computational resources needed without losing performance. For instance, we achieve the same performance for LLaVA-OneVision 7B with half the frames (8 vs. 16), and, therefore, also half the visual tokens.

**Impact of $\gamma$** We report results for different values of $\gamma$ in Tab. 3. Recall that lower values of $\gamma$ trigger paneling for shorter videos. Across all tested values, panels consistently outperform the baseline. However, skipping paneling for the shortest videos yields better results, as reflected in higher average performance for larger values of $\gamma$. Interestingly, from $\gamma = 1 \times$ fps onwards, performance on short videos is slightly better than with no paneling at all, suggesting that paneling is already beneficial at relatively short videos. We note that the impact of $\gamma$ is more pronounced on datasets with lower frame rates (e.g. TimeScope at 2 fps), where long durations are reached with fewer frames.

**Impact of $\alpha$ and $\beta$.** We explore the effect of different values for $\alpha$ and $\beta$ on two benchmarks: VideoMME, which involves standard QA across various tasks, and TimeScope with needle-in-a-haystack (NIAH) questions. The results in Tab. 4 reveal several trends. First, paneling itself improves performance overall in the majority of cases. However, increasing both $\alpha$ and $\beta$ has a clear trade-off: it provides more frames to the VLM, which benefits benchmarks involving NIAH queries, but comes at the cost of reduced spatial resolution, which hurts tasks requiring fine-grained visual detail. Second, we find that uneven paneling, i.e. when $\alpha \neq \beta$, leads to worse results than when they are equal. As such, they can be considered as a single hyperparameter.

| | VMME | | | | TIMESCOPE | |
|---|---|---|---|---|---|---|
| | short | medium | long | overall | Short | Long |
| No panels | 70.1 | 56.7 | 48.8 | 58.5 | 58.7 | 30.2 |
| $\gamma = 0$ | 69.2 | **56.8** | **49.7** | 58.6 | **70.5** | 33.8 |
| $\gamma = 0.5 \times$fps | 70.1 | **56.8** | **49.7** | **58.9** | 70.2 | 33.8 |
| $\gamma = 1 \times$fps | **70.2** | **56.8** | **49.7** | **58.9** | 69.5 | 33.8 |
| $\gamma = 2 \times$fps | **70.2** | **56.8** | **49.7** | **58.9** | 69.2 | 33.8 |
| $\gamma = 3 \times$fps | **70.2** | 56.7 | **49.7** | **58.9** | 67.7 | 33.8 |

Table 3: **Effect of the FPS constraint $\gamma$.** We report results on VMME using LLaVA-OV 7B. Results are better when not paneling on the shortest videos, but performance is robust to values of $\gamma$. The configuration used in our experiments is highlighted for clarity.

| | VMME | | | | TIMESCOPE | |
|---|---|---|---|---|---|---|
| | short | medium | long | overall | Short | Long |
| $1\times1$ | 70.1 | 56.7 | 48.8 | 58.5 | 58.7 | 30.2 |
| $1\times2$ | 69.9 | **57.6** | 48.4 | 58.6 | 65.9 | 31.3 |
| $2\times1$ | **70.9** | 55.4 | 48.1 | 58.1 | 63.5 | 32.7 |
| $2\times2$ | 70.1 | 56.3 | **50.3** | **58.9** | 69.5 | 33.8 |
| $3\times3$ | 70.1 | 56.1 | 49.9 | 58.7 | **76.5** | 33.8 |
| $4\times4$ | 70.1 | 56.4 | 48.8 | 58.4 | 73.9 | 30.9 |

Table 4: **Effect of the number of panels.** We report results on VMME using LLaVA-OV 7B. 2x2 panels achieve the best results, especially on long videos. The configuration used in our experiments is highlighted for clarity.

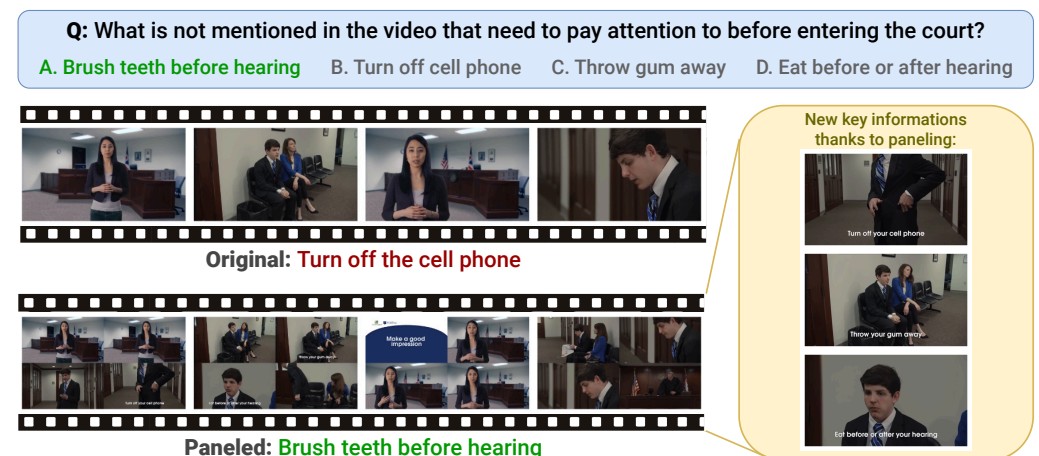

Figure 5: **Qualitative example on VideoMME.** We use LLaVA-OV 7B as the VLM. Without panels, the relevant information to answer the question is absent.

## 4.6 QUALITATIVE RESULTS

Paneling provides VLMs with a broader temporal context, allowing them to access more information. In the example of Fig. 5, the key evidence for answering the question lies in the text appearing within the video frames. Despite the lower spatial resolution of each paneled frame, the VLM successfully identifies and interprets this information, demonstrating that enhancing temporal coverage at the cost of some spatial detail is beneficial for long-video understanding.

## 5 CONCLUSION

We presented the first visual prompting strategy for long-video understanding. By combining multiple frames into panels within a single image, our method enhances the temporal resolution of existing VLMs. This training-free, model-agnostic approach can be integrated seamlessly without modifying the underlying architecture. Extensive experiments demonstrate that it consistently improves performance across a wide range of benchmarks and model designs. Furthermore, we showed that fine-tuning can provide additional gains, both compared to zero-shot paneling performance, as well as fine-tuning with the normal data representation. While our approach does not improve the understanding capabilities of the underlying VLMs, it raises the bar for new models in long video understanding to surpass before being able to justify their additional complexity.

## 6 REPRODUCIBILITY STATEMENT

We will make our code available upon acceptance. For all experiments, public available models and datasets have been used. We did not use any data for training that is not public available.

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

# A APPENDIX

# B DATASET DETAILS

- **VideoMME** (Fu et al., 2025) (VMME). A multi-modal evaluation benchmark that spans across 6 different domains (knowledge, Film & Television, Sports Competition, Artistic Performance, Life Record, and Multilingual). Video durations range between 11 seconds to 1 hour, and are categorized as *short* videos (80 seconds on average), *medium* (500 seconds) and *long* (2500 seconds).

- **TimeScope** (Zohar et al., 2025) inserts multiple short video clips ("needles") into long videos. These needles contain the key information required to answer the questions. As opposed to the usual NIAH evaluation, this forces models to understand the whole video. Videos are divided into 13 different lengths, ranging from 60 to 36, 000 seconds. We divide the evaluation into `short` (up to 3 hours, average 2590 seconds) and `long` (videos lasting 5, 8, or 10 hours, average 27, 600).

- **MLVU** (Zhou et al., 2025) covers a wide range of video lengths, from 3 minutes to 2 hours, with an average length of 15 minutes. It includes both real-world videos (e.g. egocentric, movies) and simulated videos (e.g. video games, cartoons). The benchmark provides multiple choice and open-ended generation QA tasks across 9 different categories. We evaluate on the *dev* set, which consists of 2593 QA pairs over 1730 videos.

- **MF2** (Zaranis et al., 2025) evaluates comprehension and recall of key narrative information from full-length movies (50-170 minutes). It includes 53 complete movies with an average duration of 88.3 minutes. Unlike other benchmarks, the task is to discriminate between true and false claims across 850 claim-pairs. The claims span five categories: character motivations and emotions, memorable moments, casual chains and event order.

- **VNBench** (Zhao et al., 2024) (VNB) targets three aspects of video understanding: temporal perception, chronological ordering and spatio-temporal coherence. It includes tasks such as retrieval, ordering and counting, and consists of 1350 samples ranging from 10 to 180 seconds, collected from 150 videos.

**Prompt.** Our main results show that existing VLMs are already capable of interpreting the paneled images without any additional information in the prompt. Nonetheless, adding additional directions in the textual prompt can, in some cases, improve results further.

To illustrate, we show results on LLaVA-OV 7B and Qwen2.5-VL 7B with three different prompts:

- **Prompt 1:** "You are given a sequence of images. Each image is a composite grid of video frames arranged in temporal order: panels are ordered from left to right, then top to bottom — like reading a book. Within each composite, the panels represent consecutive frames from the video. Across the sequence, the composites are shown in chronological order. When answering, interpret the full temporal sequence, not individual panels in isolation." added before the question.

- **Prompt 2:** "When answering, treat the panels as frames from one video, in order from left to right, then top to bottom." added before the question.

- **Prompt 3:** "Each image is divided into {r} rows and {c} columns of panels. Read them in left-to-right top-to-bottom order as consecutive video frames. Answer with the option's letter from the given choices directly." added after the questions. {r} and {c} are replaced by the number of rows and columns.

As can be seen from Tab. 5, even between these models, there is no consistently best prompt, but with model-specific prompts, performance can get another boost, such as Prompt 1 for VMME with LLava-OneVision, and Prompt 3 for VMME with Qwen2.5-VL. Therefore, if a user is set on a specific model, a small search over possible prompts is beneficial.

|  | No prompt | Prompt 1 | Prompt 2 | Prompt 3 |
|---|---|---|---|---|
| LLaVA-OV 7B | 58.9 | 60.1 | 59.4 | 58.8 |
| Qwen2.5-VL | 62.4 | 61.9 | 61.8 | 62.9 |

Table 5: **Effect of changing the prompt.** We report results of using additional prompts on VMME using LLaVA-OV 7B.

