# OpenReview forum: "Video Panels for Long Video Understanding"
_ICLR.cc/2026/Conference — ICLR 2026 Conference Withdrawn Submission_

### Official Review · Reviewer_9wH3 · 2025-10-30

**Soundness:** 2
**Presentation:** 3
**Contribution:** 1
**Rating:** 2
**Confidence:** 4

**Summary:**

This paper introduces a training-free visual prompting strategy for long-video understanding that trades spatial resolution for increased temporal context by compositing multiple frames into a single image, significantly boosting the performance of existing Video-Language Models (VLMs) without any modification.

**Strengths:**

- The proposed method is training-free, parameter-free, and model-agnostic, making it an exceptionally practical and low-cost solution that can be immediately applied to a wide range of existing VLMs to unlock significant performance gains on long videos, as robustly demonstrated across diverse models and benchmarks.

- The paper is well-written and easy to follow.

**Weaknesses:**

1. This work increases the number of frames within a limited context window by reducing spatial resolution, achieved by merging multiple frames into a single image. However, it is unclear why this specific implementation is necessary, as alternative methods like Slow-Fast LLaVA [1] and progressive pooling [2] also input more frames by reducing resolution. What is the distinct advantage of the proposed approach compared to these existing techniques?

[1] SlowFast-LLaVA: A Strong Training-Free Baseline for Video Large Language Models

[2] Visual Context Window Extension: A New Perspective for Long Video Understanding

2. The core idea of merging multiple video frames into a single composite image for video understanding is nearly identical to the method presented in [1]. The authors should discuss this highly relevant prior work and clearly articulate the novelty of their approach in relation to it.

[1] An Image Grid Can Be Worth a Video: Zero-shot Video Question Answering Using a VLM

**Questions:**

See Weaknesses

---

### Official Review · Reviewer_vug4 · 2025-10-31

**Soundness:** 2
**Presentation:** 3
**Contribution:** 2
**Rating:** 2
**Confidence:** 3

**Summary:**

The paper proposes a training-free, model-agnostic “visual prompting” strategy for long-video understanding: multiple frames are tiled into α×β composite “panels” and fed as a sequence of panel images. The intent is to trade per-frame spatial fidelity for greater temporal coverage under fixed vision-token/context budgets. The submission reports consistent gains on several long-video QA benchmarks and across multiple VLM families, with ablations on the panel grid and a simple fps-gating rule. While the idea is practical and easy to deploy, the evidence that it improves temporal reasoning—rather than merely exposing more frames—is not yet compelling relative to 2024–2025 standards for long-video evaluation and SOTA baselines (e.g., Video-MME, MLVU, VRBench, LongVA, Video-XL).

**Strengths:**

1. Simplicity & universality. Pure input-level transformation; plug-and-play with off-the-shelf video-VLMs.

2. Deployment-friendly. No training or architectural change required; a reasonable baseline for resource-constrained settings. Multiple models/datasets reported; clear knobs (α, β, fps-gate) and latency benefits in principle.

**Weaknesses:**

1. (Major) Method is overlooking the core challenge of long-video understanding. Packing frames into panels does not jointly preserve local detail and global, temporally grounded structure. As videos grow longer, this worsens: fine text/objects/actions are down-sampled while temporal relations remain weakly modeled (a “bag-of-frames” effect). Therefore, the contribution is of limited significance for genuine long-video understanding.

2. Limited discussion of prior “super-image/mosaic” framing. Composing multiple frames into one image (“super-image”) is known in video processing tasks; the submission should acknowledge and differentiate from this antecedent line.

3. Fine-detail failure modes under-analyzed. A discussion is absent though it is crucial to justify the spatial-for-temporal trade for long video understanding.

**Questions:**

1. Robustness to fps/shot structure. The fps-gating rule appears brittle; how does performance change under variable-fps content and uneven “needle density” (MLVU-Needle)?

2. Compatibility with long-context architectures. This method is a practical and deployable baseline that often improves accuracy by increasing temporal exposure. However, it does not, in my view, address the fundamental LVU problem—reconciling global narrative with fine spatial detail under a strict temporal model—and the current evaluation/ablations do not convincingly demonstrate gains on key long-video tasks over SOTA. Designing more task-faithful experiments and head-to-head SOTA comparisons with similar methods would substantially strengthen the contribution. Additional discussion clarifying the method’s novelty would also help.

3. Where does paneling hurt? Please provide a principled failure analysis (e.g., OCR, tiny objects, fast local motion).

---

### Official Review · Reviewer_Jhm6 · 2025-10-31

**Soundness:** 3
**Presentation:** 2
**Contribution:** 2
**Rating:** 6
**Confidence:** 4

**Summary:**

This paper introduces a simple visual prompting strategy called video panels to enhance long‑video understanding in vision‑language models (VLMs). By combining multiple frames into a single composite image, the method trades spatial detail for greater temporal coverage. The authors evaluate their technique across five video question‑answering benchmarks and multiple VLM architectures, showing consistent accuracy improvements, sometimes up to 19.4% on very long videos. They also explore fine‑tuning and compare paneling with a low‑resolution token‑pooling baseline, finding that panels generally yield better results.

**Strengths:**

The following are the strengths of the paper:

1. The proposed video panel approach is training‑free and model‑agnostic, allowing easy integration into existing VLMs.

2. The technique consistently improves performance across multiple long video understanding benchmarks and various models.

3. Combining frames into panels effectively increases temporal coverage at the expense of some spatial information, maintaining a practical balance for long videos where the spatial information is repeated over many frames.

**Weaknesses:**

The following are the weaknesses of the paper:

1. The video panel approach will not work well with native long‑video models like Qwen2.5‑VL, which already incorporate dynamic FPS sampling and absolute time encoding.

2. The impact of training models on low‑resolution pooled inputs is not explored, leaving open whether pooling could match or exceed paneling.

3. Panels provide no explicit ordering or time encoding, making it hard to answer precise temporal questions and potentially confusing frame order.

4. The technique sacrifices spatial resolution, so it is unsuitable for tasks that require fine spatial detail or small‑object recognition.

**Questions:**

The authors are encouraged to answer the following questions.

1. How is the video panel approach applied to Qwen2.5-VL which natively supports videos? Are we using it in interleave setting instead where a video can be considered as a sequence of frames?

2. Is there a way to add lightweight temporal encoding to the panels to address questions about exact timing or event order?

---

### Official Review · Reviewer_GR42 · 2025-11-03

**Soundness:** 2
**Presentation:** 2
**Contribution:** 2
**Rating:** 4
**Confidence:** 5

**Summary:**

The paper proposes a visual prompting strategy for long-video understanding: instead of changing or fine-tuning video–language models (VLMs), they pack multiple frames into comic-style “panels” within a single image, trading spatial detail for much denser temporal coverage under a fixed context window. The method is training-free, parameter-free, and model-agnostic, and they show consistent gains across five long-video QA benchmarks (VideoMME, TimeScope, MLVU, MF2, VNBench) and seven VLMs, from small (up to 16 frames) to long-context (180 frames) models.

**Strengths:**

The paper is well-written and easy to read.

Thorough experimental evaluation and ablations are provided.

**Weaknesses:**

My major concern is that the idea of shrinking the image size and putting multiple images into one frame is not novel. In fact, many prior works have tried this method, and many more effective frame sampling, compression, merging, etc have been tested on long video understanding.

For example:

Song, E., Chai, W., Wang, G., Zhang, Y., Zhou, H., Wu, F., ... & Wang, G. (2024). Moviechat: From dense token to sparse memory for long video understanding. In Proceedings of the IEEE/CVF Conference on Computer Vision and Pattern Recognition (pp. 18221-18232).

Jin, P., Takanobu, R., Zhang, W., Cao, X., & Yuan, L. (2024). Chat-univi: Unified visual representation empowers large language models with image and video understanding. In Proceedings of the IEEE/CVF Conference on Computer Vision and Pattern Recognition (pp. 13700-13710).

**Questions:**

Please refer to the weakness

---

### Note · Authors · 2025-11-14

I have read and agree with the venue's withdrawal policy on behalf of myself and my co-authors.